# Do Sleep Disorders and Western Diet Influence Psoriasis? A Scoping Review

**DOI:** 10.3390/nu14204324

**Published:** 2022-10-16

**Authors:** Ilaria Controne, Egeria Scoditti, Alessandra Buja, Alessia Pacifico, Khalaf Kridin, Massimo Del Fabbro, Sergio Garbarino, Giovanni Damiani

**Affiliations:** 1Clinical Dermatology, IRCCS Istituto Ortopedico Galeazzi, 20161 Milan, Italy; 2Institute of Clinical Physiology, National Research Council (CNR), 73100 Lecce, Italy; 3Department of Cardiologic, Vascular and Thoracic Sciences, and Public Health, University of Padova, 35128 Padova, Italy; 4Clinical Dermatology Department, San Gallicano Dermatological Institute, IRCCS, 00144 Rome, Italy; 5Lübeck Institute of Experimental Dermatology, University of Lübeck, 23562 Lübeck, Germany; 6IRCCS Fondazione Ca’ Granda IRCCS Ospedale Maggiore Policlinico, 20122 Milan, Italy; 7Department of Neuroscience, Rehabilitation, Ophthalmology, Genetics, and Maternal/Child Sciences (DINOGMI), University of Genoa, 16132 Genoa, Italy; 8Department of Biomedical, Surgical and Dental Sciences, University of Milan, 20122 Milan, Italy

**Keywords:** psoriasis, sleep, sleep disorders, western diet, diet, inflammation

## Abstract

Western diet may trigger sleep disorders and vice versa, but their single and mutual effects on systemic inflammatory diseases (i.e., psoriasis) are far from being fully elucidated. At the same time, psoriatic patients display a great burden of sleep disorders and dysmetabolisms related to an unhealthy lifestyle (i.e., diet). These patients are also affected by a chronic disorder deeply modulated by environmental factors (i.e., sleep and diet) capable to influence drug-response and disease progression. Thus, we aimed to summarize the evidence in the literature that may highlight a potential link among psoriasis–diet–sleep in order to further promote a multidisciplinary approach to psoriatic patients in the scientific community.

## 1. Introduction

Currently, the World Health Organization (WHO) reports psoriasis (PsO) as one of the five diseases that drastically influence patients’ quality of life. PsO is a systemic, chronic inflammatory skin disease related to epidermal keratinocyte hyperplasia and epidermal immune cell over-activation via the interleukin (IL)-23/IL-17 axis [1]. It is one of the most common chronic inflammatory skin diseases, with a prevalence of 1–2% worldwide [2], and almost 6–11% of patients with PsO may have inflammatory arthropathy (psoriatic arthritis) [3]. The current pathogenetic hypotheses of PsO consider a trigger (i.e., trauma or infection) that destroys keratinocytes, activating both innate immunity (i.e., neutrophils) and plasmacytoid dendritic cells that promote naïve-T-cell differentiations into Th17. The pro-inflammatory microenvironment further activates keratinocytes, causing hyperproliferation (i.e., plaques) [4].

Recently, epidemiological studies claimed a link between PsO-related inflammation and a constellation of comorbidities affecting different systems, such as cardiovascular [5,6], respiratory [7,8], neurologic [9,10] or even gastrointestinal system [11]. To date, all the above-cited comorbidities may impair both the patient’s quality of life and sleep; in fact, physicians also demonstrated that PsO patients are at a higher risk than the normal population to experience sleep disorders (i.e., insomnia or obstructive sleep apnea (OSA)) [11,12,13,14], but the real pathogenetic mechanism remains obscure.

At the same time, the pathological manifestations of PsO (phenotype) are the product of the intricate interaction between genetic susceptibility [15] and environmental factors (i.e., diet or circadian rhythm) [16] (Figure 1). Several studies pointed out that in western societies, unhealthy lifestyles may deeply condition the PsO march and flare frequency; in particular, unbalanced diets (i.e., western diet) with high fat and/or high sugar [17,18], inadequate sleep [19] and insufficient physical activity [20] may be triggers. Furthermore, all these PsO triggers may converge in creating gut dysbiosis [21], not necessarily identifiable as small intestinal bacterial overgrowth (SIBO) [22], but potentially critical for PsO and psoriatic arthritis (PsA) flares.

Thus, we decided to perform a scoping review to summarize the evidence toward PsO–western diet–sleep disorders.

## 2. Evidence Acquisition

For the present scoping review, we performed a comprehensive literature search on PubMed/MEDLINE from 1949 to 1 August 2022. No language restrictions were applied and we included only original articles, research letters and short reports; conversely, we excluded reviews, systematic reviews, letters without data and editorials.

The literature search method applied was this string: (“psoriasis” OR “psoriatic disease” OR “psoriatic arthritis” OR “inflammation”) AND ((“western diet”) OR (“sleep” OR “sleep disorders” OR “sleep apnea” OR “insomnia”)). Extensive cross-referencing and article references were carefully and manually evaluated. The literature search was performed independently by two authors (Giovanni Damiani and Ilaria Controne), and in case of disagreement, the article was discussed openly with a third experienced author (Egeria Scoditti for diet and Sergio Garbarino for sleep).

The literature search produced 839 papers; then, after eliminating duplicates and the screening phase, 81 papers were summarized into thematic paragraphs in the present scoping review (for more details, see the Preferred Reporting Items for Systematic Reviews and Meta-Analyses (PRISMA)) (Figure 2).

## 3. Sleep and Psoriasis

PsO seems to affect sleep quality, and this clinical finding is also confirmed by extensive epidemiological studies [23,24,25,26,27,28,29,30,31,32,33,34,35,36,37,38,39,40,41,42,43,44,45,46,47]. Recent studies displayed that obstructive sleep apnea (OSA) patients doubled the risk of PsO and PsA compared with heathy ones [33,34,35,38]. To further confirm this trend, PsO patients were found to be at higher risk of OSA than the general population using the apnea–hypopnea index (AHI) > 5 [23]. Researchers also observed an increase in the prevalence of resting leg syndrome (RLS) in PsO, with 15.1% and 18% in PsO versus 5% and 10% in the general populations in Europe and North America, respectively. Importantly, Chiu et al. [37] indicated that the concomitant presence of OSA and PsO exposed patients to a higher risk of Major Adverse Cardiovascular Events (MACE), especially ischemic heart attack and stroke.

Sleep disorders seem to be linked to PsO in a vicious circle in which one disease increases the risk of the other, with inflammation as a pathogenic basis bi-directionally linking sleep disorders and PsO [47,48,49,50,51,52,53,54,55,56,57,58,59,60,61]. In the PsO-to-sleep disorders directionality, there are various potential mechanisms through which PsO can act on sleep. These mechanisms operate through both direct and indirect effects. The direct effects are due to cutaneous symptoms present in PsO, such as pruritus, pain, burning sensations or, in the case of PsA, inability to move [28,39,40,41,42,43,62]. Pruritus in PsO often exhibits and/or increases during the evening, since it is regulated by circadian factors—for instance, lower cortisol levels and decreased epidermal barrier function—that lower the threshold for pruritus [30,40,41,42,43,44,45,46]. In addition, the skin plays an important role in sleep initiation, as it acts on thermoregulation and the control of the body core temperature, which normally decreases in the late evening. In contrast, these mechanisms result to be altered in PsO, where the skin diminishes the ability to dissipate heat and exhibits altered thermoregulation [52]. Instead, the indirect effects through which PsO contributes to the risk of sleep disorders may be attributable to psoriasis comorbidities that share with psoriasis a common underlying inflammatory basis [48,49,50,53]. In particular, diabetes and hypertension [57,58,60,61] were shown to be associated with insomnia, while cardiovascular disease, diabetes, obesity and psychiatric disorders were associated with OSA [59]. At the same time, systemic inflammation triggered by several pathological conditions such as OSA or even insomnia may trigger PsO flares and vice versa, thus a multidisciplinary evaluation in patients with chronic inflammation should always be considered.

On the other hand, regarding psoriasis-related sleep disorders, recent studies suggested that a possible pathogenetic mechanism could be the overproduction of oxygen radicals due to the massive pro-inflammatory cytokines spill-over from the skin and other inflamed tissues [62]. Their inhibition with biologics seems to restore the oneiric dimension, by attenuating circadian rhythm dysregulation; from this perspective, also by properly treating sleep disorders, we can hypothesize that we can decrease the possibility to develop PsO, especially in patients with positive family history.

### 3.1. Pro-Inflammatory Cytokines Shared between Sleep Disorders and Psoriasis

Common inflammatory networks are involved in the pathomechanisms underlying both sleep disorders and psoriasis [56]. In particular, pro-inflammatory cytokines such as TNF-α and IL-6 play a role in the pathogenesis of psoriasis and are implicated in sleep regulation [63]. These cytokines can be secreted by peripheral immune cells and by astrocytes and microglia in the central nervous system in response to poor sleep quantity and quality during the previous night, and produce the effect of daytime sleepiness, commonly seen in OSA. On the other hand, in chronic insomnia, there is a shift in the timing of TNF and IL-6 secretion, eliciting a modification of the hypothalamus–pituitary–adrenal axis and the hypersecretion of cortisol, that plays a role in increased wakefulness. Therefore, it is well defined that sleep deprivation can affect the immunological integrity and nocturnal secretion of cytokines, increasing the risk of PsO. Hirotsu et al. [29] examined the influence of sleep loss in an animal model of PsO by measuring cytokine and stress-related hormone levels. Male adult Balb/C mice with or without PsO were subjected to 48 h of selective paradoxical sleep deprivation. Sleep deprivation enhanced the activities of serine proteases—kallikrein-5 and kallikrein-7—which led to desquamation in the skin of psoriatic groups. In addition, mice with PsO had significant increases in specific pro-inflammatory cytokines (IL-1β, IL-6 and IL-12) and decreases in the anti-inflammatory cytokine (IL-10) after sleep deprivation, which were normalized after 48 h of sleep rebound. Another cytokine in common between sleep disorders and PsO is vascular endothelial growth factor (VEGF). Its levels are elevated in patients with severe OSA and stimulate angiogenesis and inflammation in psoriatic skin [64,65]. Moreover, studies observed that both PsO and sleep deprivation are associated with reduced levels of adiponectin, an anti-inflammatory adipokine, or increased ghrelin and decreased leptin levels, an imbalance leading to a raised feeling of hunger and appetite that increases caloric intake and subsequently the risk of overweight or obesity conditions [66]. Obesity remains one of main comorbidities associated with both PsO and sleep disorders, suggesting a potential common pathomechanism.

### 3.2. Effects of Conventional and Biological Psoriasis Therapy on Sleep Disorders

Interestingly, conventional therapies may also influence sleep quality. In fact, cyclosporin improves sleep with a fast efficacy in skin lesions [67], and methotrexate is slower but also acts on joint pain in PsA patients [14]; conversely, acitretin may de-regulate the circadian rhythm, causing insomnia [68].

To date, the impact of biological treatment (or immunotherapy) on sleep outcomes in PsO patients is poorly considered in clinical research [55]. Nevertheless, Thaçi and colleagues [31] examined PsO patients with PASI greater than 10 that deserved biological therapy (etanercept) and observed them for 24 weeks, also checking sleep parameters. They concluded that by antagonizing TNF-alpha, sleep quality drastically improved in these patients, and the current results may sustain the hypothesis that systemic inflammation may trigger/elicit or even maintain sleep disorders.

Vgontzas et al. [36] further confirmed a role of TNF-alpha in the pathogenesis of OSA by treating these patients with etanercept, obtaining greater results than with continuous positive airway pressure. Interestingly, Strober et al. tested another TNF-alpha inhibitor, namely, adamilumab, for 16 weeks, finding that beside DLQI and PASI, it also improved sleep quality in PsO patients. Remarkably, no animal models are available, and the cause–effect relation between TNF-alpha inhibition and sleep remains to be further demonstrated, since DLQI and PASI improvements may condition sleep quality. Beside direct and indirect PsO-related influence on sleep quality, these data suggest the pathogenic role of inflammatory molecules in the link between PsO and sleep disorders [17,18,21,69,70,71,72]. Interestingly, no data are present in the literature about sleep quality and inhibitors of the IL-17/IL-23 pathway. Sleep quality in PsA patients is associated with the extinguishment of joint pain, CRP and disease duration, as well as, in PsO patients, cutaneous severity, duration and patient age [14]. Thus, anti-psoriatic drug efficacy is the main sleep modulator.

## 4. Western Diet and Psoriasis

Recently, the western diet has started to be regarded as a prominent modulator of PsO severity and even as a risk factor for its development [73,74,75]. This dietary pattern, which has spread with the industrial revolution and the Modern Age, is characterized by being rich in saturated fats, trans fatty acids (FAs) and n-6 FAs, refined carbohydrates and salt, and reduced intake of n-3 FAs and monounsaturated fatty acids (MUFAs), as well as antioxidants, due to the high intake of red meat, dairies and sugars, and low intake of vegetables and fruits [76,77]. Solid evidence links this diet to the development of metabolic diseases including obesity and type 2 diabetes, as well as atherosclerosis, neurodegeneration and cancer, through mechanisms involving the instigation of chronic inflammation, oxidative stress and alterations in gut microbiota (dysbiosis) [78]. Patients with PsO presented unbalanced dietary habits resembling the western diet, as testified by their dysmetabolism clinically manifesting in obesity, metabolic syndrome and dysplipidemia [79]. This dietary habit was directly associated with the increased cardiometabolic risk profile, inflammatory markers and clinical severity of PsO [79]. On the contrary, PsO patients showed lower adherence to the Mediterranean diet, a popular and effective anti-oxidant diet, which was inversely correlated with inflammatory markers and PsO severity [80]. Since none of the single food components show to exert specific effects on the pathogenesis of PsO, it is reasonable to sustain that the biological anti-psoriatic effect is exerted by the food pattern, in other words, by the diet.

However, single foods in a diet should be carefully chosen in terms of nutrient richness and quality; for example, dietary lipids are essential to maintain cutaneous homeostasis and modulate skin immune and endocrine systems [80,81]. The composition of fatty acids (FAs) in dietary lipids significantly differs among dietary patterns. In the western diet, there is a very high intake of calories derived from fried products, butter and processed meat to the disadvantage of fish, nuts, fruits and vegetables. As such, saturated fatty acid (SFA) intake is elevated through the consumption of meat, butter and palm oil, while the intake of n-3 PUFAs, such as α-linolenic acid (18:3), eicosapentaenoic acid (EPA; 20:5) and docosahexaenoic acid (DHA; 22:6), which can be found in fish and nuts, is low. This fatty acid profile contributes to the negative health outcomes associated with the western diet by increasing the risk of dyslipidemia, obesity, diabetes or cardiovascular diseases, as well as total mortality [82].

Dietary SFAs represent a major risk factor for PsO exacerbation, even independently of obesity. In PsO patients, the serum levels of free FAs were associated with disease severity [83]. Furthermore, by introducing SFAs in the diet elicited a psoriatic flare in mice, suggesting the prominent pro-inflammatory role exerted by the western diet [83].

Interestingly, n-3 PUFAs play a key anti-inflammatory role in rodents (i.e., mice and rats), as well as in humans. Interestingly, PASI inversely correlates with the serum level of n-3 PUFA, and the SFA/unsaturated FA ratio increases with the duration of the disease [84].

### 4.1. Effects of Western Diet and Psoriasis on Microbiota

The perturbation of gut and/or skin microbiota may trigger systemic inflammation or even a flare of a pre-existent inflammatory condition (i.e., PsO) through pathobiont colonies increase [18,71,72,85,86]. During cutaneous inflammation, antimicrobial peptide release becomes less effective, and the interaction between bacteria and immune system, more frequent, thus acting as inflammatory triggers (i.e., *Staphylococcus aureus*) [87]. PsO patients exhibited a depletion of *Corynebacterium* spp., *Lactobacillus* spp., *Burkholderis* spp. and *Propionibacterium acnes* in cutaneous microbiota, as well as *Faecalibacterium prausnitzii* and *Akkermansia muciniphila* in gut microbiota. Conversely, Firmicutes and Actinobacteria spp. proliferated in the gut mucosa, reducing the pool of microbes capable of producing short chain fatty acids (SCFAs) [88,89,90]. Thus, microbiota represent the living, dynamic filter of dietetic nutrients, and its unbalance may have pro-inflammatory effects. This idea received a proof of concept with the clinical study by Deng et al., in which they described the beneficial role of specific probiotics (*Bifidobacterium infantis 35,624* and *Lactobacillus pentosus GMNL-77*) in decreasing imiquimod(IMQ)-induced psoriasiform eczema and its systemic inflammation (i.e., TNF-α and IL-6) [91,92,93].

Several studies suggested the influence of environmental factors such as dietary composition and, in particular, the western diet on microbial community and function [94,95,96]. The western diet was associated with intestinal barrier disruption and gut dysbiosis with an altered profile of bacterially produced metabolites, resulting in metabolic endotoxemia, immune system deregulation and systemic inflammation [78,97].

Since the IL-17/Il-23 pathway represents a bridge between innate and adaptive immunity against microbes, its modulation by nutrients is of particular interest. With specific regards to psoriasis, in a recent study in an IL-23–mediated model of PsO and PsA, Shi et al. revealed that a short-term western diet intake exacerbated both intestinal dysbiosis as well as psoriasis-like skin and joint inflammation [18]. Assuming that diet and inflammation may influence gut dysbiosis, they proved that, by switching from the western to a normal diet or even treating with broad spectrum antibiotics, IL-23-induced skin and joint inflammation mitigated. Furthermore, fecal microbiota transplantation from western-diet-fed donors into mice pretreated with broad-spectrum antibiotics revealed that gut microbiota triggered γδ T-cell infiltration into the dermis. Thus, microbiota modulation may be the key to also improve drug response in psoriatic patients.

The high-fat diet (HFD), as well as the western diet, was associated with lower microbial production of SCFAs, including butyrate, propionate and acetate, which are fermentation products of dietary fibers produced in the colon and are able to exert systemic effects, including at the skin level. Interestingly, SCFAs can be also produced by commensal bacteria in the skin [97]. A protective role of SCFAs against PsO was shown by several lines of evidence. SCFAs are able to (1) promote T-reg differentiation, activation and function; (2) inhibit the intestinal dendritic-cell production of IL-23 while inducing the expression of anti-inflammatory genes; (3) reduce skin inflammation in a mouse model of PsO, as well as downregulate IL-17 expression, and induce IL-10 and Foxp3 expression in animal and human psoriatic skin lesions [98].

### 4.2. How Lipids Influence Immune System Responses

#### 4.2.1. Inflammasomes

Several studies demonstrated that macronutrients typical of the western diet, including SFAs, are able to induce and/or amplify pro-inflammatory responses involved in PsO development and progression [99,100,101]. Interestingly, IMQ-treated mice subjected to a HFD presented more severe clinical and histological (micro-abscesses and scaling) PsO than the ones following a regular diet. The HFD was responsible for the pathological activation of the nucleotide-binding domain, leucine-rich repeats containing family, pyrin domain-containing-3 (NLRP3) inflammasome. Inflammasomes are cytoplasmic multiprotein complexes for intracellular signaling that activate IL-1β via caspase-1, in response to different triggers (i.e., trauma or infections). Inflammasome activation was linked to the pathogenesis of metabolic and cardiovascular disease and is genetically associated with PsO [102]. Recent findings reported evidence of NLRP3 inflammasome activation in peripheral blood cells in PsO patients, in parallel with increased caspase-1 reactivity and serum levels of inflammasome-generated IL-1β and IL-18 [103]. TNF-α inhibitors are capable to turn off the inflammasome, as testified by the decreased plasmatic levels of IL-1β [103], thus suggesting a role of TNF-α-mediated NLRP3 inflammasome activation in patients with PsO and its contribution to systemic inflammation. In IMQ mice, the HFD, but not the regular diet, increased the expression of activated caspase-1 and IL-1β in the skin. The HFD is also a strong activator of the IL-17 pathway, as testified by higher levels of IL-17A in both the dermis and serum of IMQ-treated mice. Concordantly, SFAs were reported to activate inflammasomes and subsequent IL-1β release in macrophages [104], thus possibly mediating the exacerbation of psoriatic dermatitis. In addition, SFAs are powerful activators of myeloid residential cells (i.e., plasmacytoid dendritic cells) capable of further activating keratinocytes by boosting the psoriasis-related inflammation signal [83]. This suggest that dietary lipids are involved in the development and progression of psoriasis via systemic inflammation and inflammasome activation.

To prove the concept, Christ and colleagues fed LDLR^−^/^−^ mice with the western diet and then shifted to a chow diet (pelleted obesogenic diet), finding that the western diet increased systemic inflammation and promoted a pro-inflammatory imprinting on granulocyte monocyte precursor cells (GMPs); the effects were mitigated when mice abandoned the western diet in favor of the chow diet [105]. In addition, the western diet caused dysregulation in bile acids synthesis and release, further amplifying lipid dysmetabolism [105].

#### 4.2.2. Adipokines, Cytokines and Chemokines

Adipose tissue (AT) dysfunction is currently regarded as a trigger for several inflammatory conditions, both local and systemic, such as PsO, and can be further amplified in obese patients. AT is still regarded as an endocrine organ secreting FAs and a panel of cytokines/chemokines and adipokines, such as leptin, resistin and adiponectin. In conditions of obesity or under chronic inflammation, dysfunctional adipocytes change their secretory profile toward a more pro-inflammatory state that induces the cutaneous infiltration and pro-inflammatory activation of immune cells (e.g. macrophages, neutrophils, lymphocytes). The resulting low-grade inflammation state in obesity is a central pathogenetic moment is PsO development [106]. Several studies [6,107,108] observed that the serum level of leptin is particularly elevated in obese individuals and associated with leptin resistance [109].

Leptin displays a pleiotropic function, spacing from appetite modulation to acting as a pro-inflammatory mediator capable of sustaining Th-1 differentiation and T-reg inhibition [110]. Therefore, PsO patients subjected to high-fat diets have a higher risk of obesity and to display higher levels of leptin [111].

Conversely, Lihn et al. revealed that adiponectin exerted an anti-inflammatory effect opposed to leptin capable of inactivating the psoriatic microenvironment, restoring cutaneous homeostasis [112,113]. Since the blood adiponectin levels resulted low in both obese people and PsO patients, adiponectin regulation is a potential mechanism mediating the relationship between PsO and AT [113].

Remarkably, antimicrobial peptides (AMPs) may also enter in the adipokines, since during adipogenesis, as well as injuries, they are produced and released [114]. Among AMPs, there is also cathelicidin antimicrobial peptide LL-37, capable of binding double-strained DNA filaments both exogenously and endogenously, creating the dangerous contact between self-DNA and immune system, thus starting PsO [115]. Cytokines IL-17A and IL-22 seem to play a crucial role in the development of PsO. Kanemaru et al. examined the mRNA expression of these cytokines in an obese mouse model with IMQ-induced psoriasiform dermatitis, finding five times more mRNA related to these cytokines than in the controls, suggesting a prominent AT-triggered activation of the IL-17 pathway. Additionally, it is interesting to note that food intake restriction partially decreased cytokine production in obese mice. Furthermore, a high-fat dietary pattern and IMQ treatment together stimulated the production of Reg3γ (regenerating islet-derived 3γ), an antimicrobial protein critical in psoriatic epidermal hyperplasia, since it is able to activate phosphatidylinositol 3-kinase (PI3K) in keratinocytes [116]. In particular, AT also releases palmitic acid, which is capable of influencing in vivo and in vitro Th-17 differentiation, infiltration and migration to lymph nodes [117]. Interestingly, a HFD further enforced palmitic acid release and this fact may explain the harmful effect of a HFD on PsO patients [118].

#### 4.2.3. n-6 PUFA-Derived Prostanoids and Leukotrienes

Prostanoids and leukotrienes (LTs) are two well-known pro-inflammatory n-6 long-chain PUFAs (20:4) derived, produced and released during traumas or in response to inflammatory cytokines. They also contribute to create and maintain the inflammatory microenvironment in PsO, but their potential pathogenetic role is far from being fully elucidated [119]. Ueharaguchi and colleagues performed a targeted lipidomics on IMQ-treated mouse skin, finding that thromboxane (Tx) A2 synthase was consistently upregulated; so, lipid mediator TxA2 increased in parallel with dermatitis [120]. Moreover, they used TXA-receptor-deficient mice and treated them with IMQ, observing a significant decrease in cutaneous infiltrating γδ Th17. The treatment of wild-type mice with an inhibitor for TXA2 synthase was also effective in reducing IL-17 production and disease severity. Prostaglandin (PG) E2 is another prostanoid that seems to be able to increase psoriatic dermatitis via the regulation of the IL-23/IL-17 pathway. Since Schirmer et al. demonstrated in vitro that dendritic cells may increase IL-23 production upon the fibroblastic release of PGE2 [121], Lee et al. showed in vivo and in vitro that Th-17 cells used PGE2 as an autocrine pro-proliferative mediator to maintain the pro-inflammatory microenvironment typical of the psoriasis plaque [122]. Since nonsteroidal anti-inflammatory drugs inhibit COX activity, increasing the production of LTs and inducing the progression of PsO, LTs are supposed to be additional disease-promoting factors in PsO. In particular, Sumida et al. described the importance of LTB4-BLT1 (LTB4 high affinity-receptor) signaling in a IMQ-induced PsO model. In fact, the inhibition of LTB4 synthesis or the genetic deficiency of BLT1 attenuated neutrophil infiltration in the skin and improved the symptoms of psoriasis [123]. Despite dedicated clinical trials with leukotrien inhibitors did not sort any significant anti-psoriatic effects, in vitro leukotriens instructed and increased neutrophils infiltration via LTB4–BLT1 signaling [124].

#### 4.2.4. EPAs and DHAs (n-3 PUFAs)

Surprisingly, n-3 long-chain PUFAs may also exert a significant anti-inflammatory role in the pathogenesis of PsO, and obviously, they are regarded as anti-psoriatic drugs in line with conventional and biological therapies. For this reason, the anti-psoriatic effects of n-3 PUFAs are being studied, and the specific mechanisms by which they are involved are now to be well defined. n-3 PUFAs display a biological anti-inflammatory action directly by competing for the arachidonic acid present in the membrane and indirectly with its metabolites, which contribute globally to inactivate the inflammatory microenvironment typical of psoriasis [125]. Interestingly, Qin et al. utilized fat-1 transgenic mice that had the characteristic to convert n-6 PUFAs into n-3 PUFAs and compared them with the wild type after IMQ cream treatment. Fat-1 mice displayed less Th-17 and a lower rate of Th17/T-reg in the spleen, demonstrating the crucial role of n-3 PUFAs in modulating the Il-17 pathway [126]. Similarly, the relative systemic cytokine spill over was also limited. Allen et al. suggested that EPA contributes to increase IL-6 receptors on the CD4+ T-cell surface via STAT3 [127]. Kong et al. in their study revealed that DHA-treated dendritic cells showed a reduced ability to induce Th17 differentiation and proliferation [128]. Moreover, recent evidence found that resolvin E1 (RvE1), a DHA metabolite, inhibited Leukotriene B4 receptor 1 (BLT1) and modulated the dendritic-cell capability to present antigens and γδ T-cell pro-inflammatory effects [129,130,131]. Likewise, resolvin D1, another DHA metabolite, reduced IMQ-induced PsO by acting on Nuclear factor kappa-light-chain-enhancer of activated B cells (NF-κB), a prominent inflammatory pathway involved in PsO initiation and maintenance [132].

## 5. Sleep and Dietary Patterns: An Integrated View

A bidirectional relationship does exist between diet and sleep/sleep disorders [133]. In particular, sleep disorders or mistimed sleep are linked to increased body mass index (BMI), overfeeding and irregular eating patterns, as well as unhealthy eating behaviors [134,135,136,137]. Indeed, both single nutrients and food patterns influence the gut-associated lymphoid tissue (GALT) and melatoninergic ones. These changes in hormone levels may increase appetite and thus influence dietary intake, predisposing to weight gain and obesity [138,139,140]. Moreover, the fatigue and excessive daytime sleepiness associated with poor quality sleep restrict the possibility of physical activity and also lead to a compensatory rise in caloric intake [141]. Interestingly, patients with sleep disorders displayed central obesity, but the insomnia/OSA group showed higher adherence to the western diet [142,143,144,145]. On the other hand, unhealthy diets such as the western diet negatively influence sleep parameters, thus leading to sleep disturbances or disorders [146,147], through potential mechanisms including body weight gain and overweight/obesity, related metabolic and vascular diseases, pro-inflammatory and pro-oxidant action, decreased synthesis of melatonin, induction of gut dysbiosis and altered pattern of bacterially produced metabolites, which can include sleep regulators such as SCFAs, GABAs, glutamate and serotonin [148]. Interestingly, the perturbation of the circadian rhythm, which modulates both sleep and appetite, with intermittent circadian fasting (Ramadan fasting) displayed surprising anti-inflammatory effects in both dermatological [149,150,151] and rheumatic diseases [152], included PsO [153].

Contrarily, a healthy diet (i.e., Mediterranean one) was associated with a correct sleep behavior (quality and quantity), through mechanisms going beyond the recognized benefit of weight reduction [154]. Campanini and colleagues affirmed that the adherence to the Mediterranean diet is protective in terms of both preventing and improving sleep disorders [136,137].

Nowadays, literature records seem to claim a link between sleep disorders and inflammation, so sleep may benefit anti-inflammatory therapies in patients with IMIDs. At the same time, pharmacological interventions should be provided in synergy with a healthy lifestyle (i.e., Mediterranean diet) to maximize their efficacy. In fact, the Mediterranean diet has a recognized anti-inflammatory effect related to food compounds (i.e., MUFAs and n-3 PUFAs, resveratrol and other polyphenols) and food quality/quantity. The Mediterranean diet (i.e., lemons and tomatoes) contains melatonin, which further implements the endogenously produced one and contributes to maintaining and regulating sleep. Furthermore, melatonin is currently also supplemented for sleep disorders but has never been tested as a complementary therapy in PsO. Thus, a health dietary pattern can be helpful in the management of sleep disorders, but more clinical studies are necessary to validate this hypothesis [155,156]. Sleep disorders and the western diet are bidirectionally linked and synergically promote systemic inflammation by de-programming human lipid metabolism, potentially triggering also PsO [12]. Further human studies are warranted to specifically evaluate the presence of sleep disturbances and the diet in PsO patients, and their relative and cumulative contribution to PsO development and clinical course.

## 6. Conclusions

Sleep disorders and an unhealthy lifestyle, such as the western dietary pattern, negatively impinge on skin homeostasis and regulate or exacerbate the development, clinical course and outcomes of PsO [157,158]. Indeed, both factors and PsO disease have in common an alteration of immune-mediated responses and chronic inflammatory conditions [159,160]. Remarkably, exposure to other factors, such as alcohol, infections, vaccines, pollution and smoking, may also influence the patient course and modulate sleep quality, so together with the diet, clinicians should advocate a change in the patient’s lifestyle [161,162,163,164,165,166]. As such, it is important for dermatologists and practitioners to consider a screening for both sleep disturbances and lifestyle factors such as the diet, which are tightly intertwined, often disregarded and never considered simultaneously, in the management of PsO disease. These considerations could be useful for ameliorating the standard clinical care for PsO patients [167].

## Figures and Tables

**Figure 1 nutrients-14-04324-f001:**
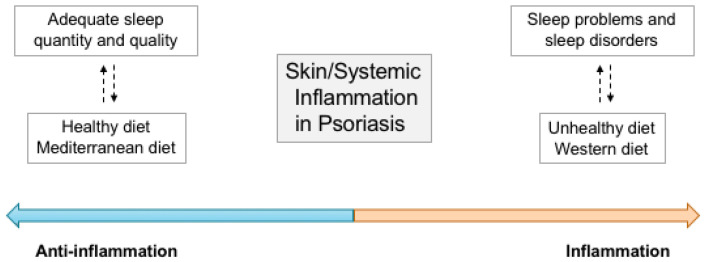
Psoriasis is associated with sleep disorders and unhealthy dietary patterns through the activation and/or regulation of systemic and skin inflammation. On the left, the scheme shows protective effects obtainable via adequate sleep quantity and quality together with healthy diet on psoriasis disease.

**Figure 2 nutrients-14-04324-f002:**
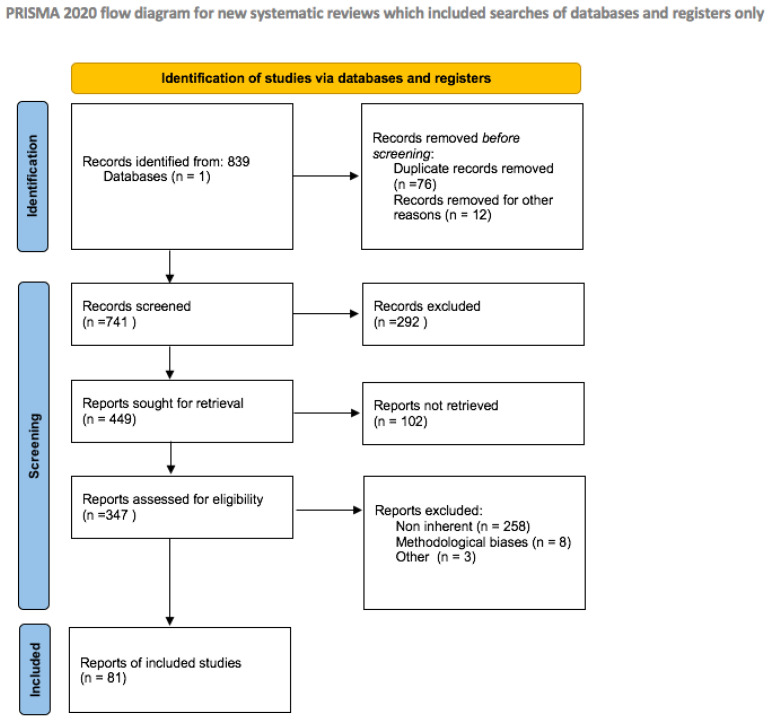
PRISMA flow diagram with a detailed report of the literature revision.

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
