# Peer review of "Do Sleep Disorders and Western Diet Influence Psoriasis? A Scoping Review"

_nutrients, 2022, doi:10.3390/nu14204324_

Round 1
Reviewer 1 Report
1. it would be grateful if authors could add a PRISMA diagram.
2. In the section 3.2. please not only describe the effect of TNF inhibitors, but also IL-12/23 inhibitors, IL-17 inhibitors. The authors should describe the potential effect of each inhibitor from a group of inhibitors. For example, for anti-TNF drug: adalimumab, etanercept, infliximab, golimumab, certolizumab pegol etc..
3. Please describe the effect of conventional psoriasis therapy, e.g. cyclosporine to sleep
4. Why did this paper focus on only western-diet ?
5. the authors should also describe the influence of environmental factors, such as alcohol consumption, smoking, stress etc.
6. please consider to cite https://doi.org/10.3390/life12010118
7. English needs a minor revision.
Author Response
ANSWERS TO REVIEWER 1
We really appreciated all your suggestion to improve our manuscript. We tracked all the changes made.
R1: It would be grateful if authors could add a PRISMA diagram.
A: We agree with the suggestion and we add the PRISMA diagram version 2020
R1: In the section 3.2. please not only describe the effect of TNF inhibitors, but also IL-12/23 inhibitors, IL-17 inhibitors. The authors should describe the potential effect of each inhibitor from a group of inhibitors. For example, for anti-TNF drug: adalimumab, etanercept, infliximab, golimumab, certolizumab pegol etc..
A: We thank the Reviewer 1 for this suggestion. We search the literature and we found only few evidence on other biologics, merely indirect since an interested study performed by Krajewska-Włodarczyk et colleagues sustained that sleep quality in PsA patients are related to pain tender joint count, C-reactive protein (CRP) concentration, patient's age and duration of psoriasis, conversely in PsO sleep quality is related to severity of skin lesions, duration of psoriasis and patient's age.Remarkably, the single drug did not influence directly sleep quality but only in terms of cutaneous/articular efficiency. We add the references.
R1: Please describe the effect of conventional psoriasis therapy, e.g. cyclosporine to sleep
A: We thank for the suggestion and we reported literature evidence also toward MTX, cyclosporine and acitretin in the text.
R1: Why did this paper focus on only western-diet ?
A: We decided to focus only on western diet since it is the diet more present in western countries ad actually a dedicated review is not present. Western diet was also tested in mouse models and we are dedicating a massive intervention on our patients with dietologists in our department, so this review was supposed to be a literature screening to highlight the current evidence an build new ones in the next future.
R1: The authors should also describe the influence of environmental factors, such as alcohol consumption, smoking, stress etc.
A: We thank for the insight and we add a dedicated paragraph in the text.
R1: please consider to cite https://doi.org/10.3390/life12010118
A: We thank for the citation that was greatly appreciated.
R1: English needs a minor revision.
A: We agree and we submitted the manuscript to a mother Language to perform some corrections.
Reviewer 2 Report
-
This is a well-written article that identifies an important review gap. Given the complexity involved, the author has produced many positive and welcome outcomes. The literature review offers a useful overview of current research and policy, and the resulting bibliography provides a very useful resource for current practitioners.
-
Our lifestyle affects our health and it is important for doctors to make patients aware of what they can change
Author Response
Thank you very much for your appreciation.
Lifes styles may be changed in HS patients to mild the severity of the patients' journey